# Quality of Life and Psychological Distress in Portuguese Older Individuals with Tinnitus

**DOI:** 10.3390/brainsci11070953

**Published:** 2021-07-20

**Authors:** Haúla F. Haider, Sara F. Ribeiro, Derek J. Hoare, Graça Fialho, Deborah A. Hall, Marília Antunes, Helena Caria, João Paço

**Affiliations:** 1ENT Department, Hospital Cuf Tejo—Nova Medical School, 1350-352 Lisbon, Portugal; joao.paco@cuf.pt; 2CUF Academic and Research Medical Center, 1350-352 Lisbon, Portugal; 3Comprehensive Health Research Centre (CHRC), 1169-056 Lisbon, Portugal; 4National Institute for Health Research (NIHR) Nottingham Biomedical Research Centre, Hearing Sciences, Mental Health and Clinical Neuroscience, School of Medicine, University of Nottingham, Nottingham NG7 2QL, UK; derek.hoare@nottingham.ac.uk; 5Faculty of Sciences, BioISI—Biosystems & Integrative Sciences Institute, University of Lisboa, 1749-016 Lisboa, Portugal; mdfialho@gmail.com (G.F.); mhcaria@fc.ul.pt (H.C.); 6Department of Psychology, School of Social Sciences, Heriot-Watt University Malaysia, Putrajaya 62200, Malaysia; deborah70hall@gmail.com; 7Centro de Estatística e Aplicações, Departamento de Estatística e Investigação Operacional, Faculdade de Ciências, Universidade de Lisboa, 1749-016 Lisboa, Portugal; marilia.antunes@ciencias.ulisboa.pt; 8Polytechnic Institute of Setubal, School of Health -Biomedical Sciences Department, CIIAS, 2910-761 Setubal, Portugal

**Keywords:** tinnitus, quality of life, psychological distress

## Abstract

Tinnitus is the perception of a sound without an external source, often associated with adverse psychological and emotional effects leading to impaired quality of life (QoL). The present study investigated QoL and psychological distress in tinnitus patients and analysed the effects of associated comorbidities. Tonal and speech audiometry, tinnitus assessment, and clinical interviews were obtained from 122 Portuguese individuals (aged from 55 to 75). Portuguese versions of the Brief Symptoms Inventory (BSI), the Medical Outcomes Study Short Form Health Survey (MOS SF-36) and Tinnitus Handicap Inventory (THI) were used to evaluate psychological distress, health-related QoL, social difficulties and tinnitus severity. The presence of tinnitus was significantly associated with hearing loss. The increases in tinnitus severity were associated with decreases in QoL, particularly regarding MOS SF-36 subscales “perception of health”, “social functioning”, and “mental health”. Regarding BSI, patients with greater tinnitus severity had more severe psychopathology symptoms, measured with scales “Obsessive–compulsive”, “Depression”, “Anxiety”, “Hostility” and “Phobic Anxiety”. Our study supports the notion of the negative impact of increased tinnitus severity on QoL and psychological distress in older adults. Presented data strengthen the importance of a multidisciplinary approach to tinnitus assessment and treatment.

## 1. Introduction

Tinnitus is a symptom involving the subjective perception of sound in the absence of an external source. It is a common symptom, affecting 12% to 30% of the general population, and rising with age [1,2].

The pathophysiological mechanisms underlying tinnitus remain unknown, and the clinical approach to diagnosis involves a detailed anamnesis, searching for associated factors and comorbidities. Hearing loss or hearing difficulty are the comorbidities most frequently associated with tinnitus [3], which is also one of the common chronic diseases in elderly individuals [4]. Other relevant comorbidities are history of head injury or arthritis, and current or former smoking [2]. With regards to prescription drugs, intake of the thyroid hormone levothyroxine and proton-pump inhibitors for reducing stomach acid production have also been associated with tinnitus [5]. Hyperacusis and mental health comorbidities such as anxiety and depression may be associated with hyper-vigilance and selective attention to tinnitus [6], amplifying the impact of tinnitus as well as impeding treatment.

When people are asked to report on the impact of tinnitus on their daily life, several recurring themes emerge. These were collected and categorized by Hall et al. [7] into a taxonomic framework that encompassed physical health problems (e.g., sleep difficulties, somatic complaints), psychological difficulties (e.g., distress, concentration difficulties), impaired social relations (e.g., impact on social life and work activities), and negative attributes of the tinnitus sound (e.g., loudness, intrusiveness), and significant others. These domain-level groupings map closely to the major category headings given by the World Health Organization (WHO), which correspond to a conceptual framework of QoL. QoL is a broad ranging-concept that refers to a state of physical, mental, and social well-being. General measures of QoL typically consider a person’s physical health, psychological state, level of independence, social relationships, and their relationship to their surrounding environment, and such instruments assess QoL in a generic way so that the impact of one disease can be compared directly to another (e.g., WHOQOL-BREF) [8]. The MOS SF-36 is another generic QoL instrument that has been administered for tinnitus, with mean scores on all subscales below that of the general population [9], and global scores decreasing (worsening) as a function of tinnitus severity [10,11]. In contrast, condition-specific measures of QoL consider life impacts from the perspective of a particular health condition (e.g., THI) [12]. Thus, while the items can be tailored to the specific disease of interest, the QoL scores cannot be interpreted with respect to other diseases. The THI predominantly comprises statements about psychological and emotional effects of tinnitus (68%) and impacts of tinnitus on lifestyle (20%) [12].

A generic measure of mental health and psychological distress is the BSI, and this has been administered in several tinnitus studies to assess mental health symptoms and psychological distress [11]. The BSI includes items assessing somatization, depression, and anxiety, all of which can be aggravated by tinnitus. In a recent clinical study for tinnitus and hyperacusis evaluating the benefit of a multi-modal treatment program including a Cognitive Behavioral Therapy (CBT) component, the authors reported a significant reduction on a tinnitus-specific QoL (Tinnitus Questionnaire score reduced from 35.7 to 20.3 points out of 100) and on the BSI (from 49.6 to 25.2 points out of 100) [13].

The purpose of the present study was to quantitatively investigate QoL in tinnitus older individuals presenting at a specialist hospital clinic in Portugal. Specifically, we analysed the relationship between tinnitus severity, QoL and mental health, and investigated the effects of associated comorbidities.

## 2. Materials and Methods

### 2.1. Participants

Our sample included 122 individuals, 89 of whom suffered from tinnitus, consecutively recruited from the ENT consultation of Hospital CUF Infante Santo, between 2014 and 2018.

WHO classifies aging into four stages: Middle age: 45 to 59 years; Elderly: 60 to 74 years old; Elder: 75 to 90 years old; Extreme old age: 90 years upwards. Since the focus of our study was on older individuals, we decided that the range from 55 to 75 years old would give us a good appreciation of the aging process in regards to tinnitus and related comorbidities. Our sample was gender balanced (63 women and 59 men).

Inclusion criteria were Portuguese nationality and availability to join the study. Exclusion criteria were age younger than 55 or older than 75, cognitive impairment without the capacity to understand and sign an informed consent, the presence of uncompensated medical disorder and the presence of disease of the outer ear (occlusive exostosis, outer otitis). We also excluded people with Ménière’s disease, chronic otitis media, otosclerosis, history of ototoxic drugs use, exposure to massive noise, history of previous malignancy with chemotherapy, history of autoimmune disorders, or neurodegenerative or demyelinating disease.

We included participants with subjective chronic and non-pulsatile tinnitus (duration longer than six months), so excluded objective or somatosensory tinnitus.

All participants were subject to clinical, audiological and tinnitus evaluation and answered the MOS-SF36 and the BSI questionnaires. Information concerning sociodemographic, health and daily life aspects were also collected.

According to tinnitus presence, participants were divided into the “Tinnitus group” and the “No Tinnitus group”. For some analyses, Tinnitus group individuals were categorized as either Irrelevant/Mild Tinnitus group or Moderate/Severe/Catastrophic Tinnitus group (according to THI scores less than 37 or more than 37 respectively—cut-off proposed at European Tinnitus Guidelines [14]), to assess how tinnitus severity affects QoL and/or is related to psychopathologies.

The present study was approved by the Ethical Committees of Hospital CUF Infante Santo, Nova Medical School (nº65/2014/CEFCM) and the National Department of Personal Data Protection (authorization number:1637/2016). The study was conducted in accordance with the Declaration of Helsinki.

### 2.2. Clinical Evaluation

#### 2.2.1. Audiological Assessment

We performed pure tone audiometry (air and bone) to evaluate the hearing thresholds according to ISO 8253 and 389. Evaluation was accomplished in a soundproof booth employing an Interacoustics^®^, Assens, Denmark audiometer (Model: AC40, Serial No.: 98 019 046) and TDH39/HDA300 headphones fitted with noise-excluding headset ME70 and bone conductor B-71. Standard tonal audiometry and extended high frequency audiometry were performed at frequencies from 0.25 kHz to 16 kHz.

The category of hearing loss (HL) was defined according to the recommendations of the Bureau International d’Audiophonologie (BIAP). Pure tone average (PTA) was calculated as the average of thresholds at 500 Hz, 1000 Hz, 2000 Hz and 4000 Hz. Frequencies not heard were given a value of 120 dB. For statistical analyses we have also considered high frequencies average (HFA) calculated as the average thresholds across 2, 4, and 8 kHz [15] and a very high frequencies average (VHFA), calculated as the average thresholds at frequencies 8, 10, 12, 14 and 16 kHz.

Speech test recognition (STR) was conducted with headphones using either an mp3 player or open field (avoiding the possibility of lip-reading) and recorded as the number of disyllables that each participant repeated correctly. The intelligibility threshold corresponding to 50% correct identification or more of disyllables from a standard list of two-syllable words intends to measure hearing sensitivity threshold through the intensity level identification and allows the measurement of hearing sensitivity threshold. Speech discrimination evaluates the lowest intensity level at which a listener can understand speech and was also registered.

#### 2.2.2. Tinnitus Assessment

Psychoacoustic assessment of tinnitus included information about loudness match, pitch match, minimum masking level (MML) or Feldmann masking curves, residual inhibition, and loudness discomfort levels (LDL). Tinnitus severity was evaluated with the validated Portuguese version of the THI [16,17]. THI results allow the identification of the most affected aspects of tinnitus in daily life and establish a classification of tinnitus severity from Slight or no handicap (Grade 1), Mild (Grade 2), Moderate (Grade 3), Severe (Grade 4) and Catastrophic (Grade 5).

#### 2.2.3. MOS SF-36

The Portuguese validated version [18] of the MOS SF-36 was used to evaluate health-related QoL. MOS SF-36 questions evaluate physical and mental health through the assessment of aspects related to function, well-being, disability, and personal evaluation. MOS SF-36 questions reflect eight health constructs: Physical Functioning (MOS.PF), Role-Physical (MOS.RP), Bodily Pain (MOS.BP), General Health Perceptions (MOS.GHP), Vitality (MOS.V), Social Functioning (MOS.SF), Role-Emotional (MOS.RE), and Mental Health (MOS.MH). These domains are organized into two summary scales: Physical Component Summary scale (MOS.PCS) and Mental Component Summary scale (MOS.MCS) where each domain is scored from 0 and 100, with a higher score indicating better QoL. The additional Health Change (HC) scale is an informational scale, which measures the degree of change in general in the patient’s health and is not part of the dimensions mentioned above.

#### 2.2.4. BSI

Psychological symptoms were evaluated using the Portuguese version [19] of the BSI. The BSI is a self-reported inventory, composed of 53 items, with items scaled from 0 (not at all) to 4 (extremely). Participants’ responses correspond to the psychological symptoms experienced in the last 7 days and higher scores indicate more severe psychopathology or psychological distress. The BSI includes nine subscales: somatization (SOM), obsessive–compulsive (O-C), interpersonal sensitivity (I-S), depression (DEP), anxiety (ANX), hostility (HOS), phobic anxiety (PHOB), paranoid ideation (PAR) and psychoticism (PSY). Four additional items (sleep disturbances, thoughts of death, feelings of guilt, and loss of appetite) are considered, together with BSI dimensions, to provide three global indices of distress: the General Severity Index (GSI), the Positive Symptom Distress Index (PSDI), and the Positive Symptom Total (PST). GSI assesses the level of present distress. PSDI assesses the level of suffering, and PST assesses the set of all symptoms reported [20].

### 2.3. Statistical Analysis

Descriptive statistics included counts, proportions, means, and standard deviations calculated for the collected data, considering the whole sample and also the data stratified by presence/absence of tinnitus and/or the levels of tinnitus severity. Associations between categorical variables were assessed using the chi-squared test or the Fisher exact test, depending on the variables and the data. Association between quantitative variables and MOS SF-36 and BSI scores were assessed by Pearson correlation and tested for significance. Comparisons of quantitative variables between levels of binary variables was achieved through the Welch two sample t-test. Multiple linear regression models were used to assess the effect of the presence of tinnitus on the HL threshold, controlling by age and gender. Main effects of tinnitus severity on MOS SF-36 and BSI summary scales were assessed through one-way ANOVA. As post-hoc analyses, the effects of tinnitus severity in the results of MOS SF-36 and BSI questionnaires were examined in multivariate linear regression models. In both cases, posterior analyses included pairwise comparisons adjusted by the Tukey method for comparing a family of estimates.

## 3. Results

Sociodemographic characteristics, noise related variables, past and present medication including smoking status, and comorbidities are presented in Table 1, Table 2, Table 3 and Table 4, respectively, for the No Tinnitus and Tinnitus groups. The majority of the participants reported suffering from tinnitus (*n* = 89, 73.0%). Only 33 participants (27.0%) said they did not suffer from any form of tinnitus. No statistically significant differences in age were found between groups (Welch two sample *t*-test, *p* = 0.744). Average age was 63.8 years (SD = 5.7) in whole sample, 63.5 years (SD = 5.5) in the No Tinnitus group and 63.9 years (SD= 5.8) in the Tinnitus group. No significant differences were found between groups on noise-related characteristics or comorbidities. As for medication and smoking habits, no statistically significant differences were found, except when comparing the proportion of participants under present antidepressant or anxiolytic medication (Chi-square test, *p* = 0.041) unadjusted *p*-values.

Tinnitus Annoyance and the other tinnitus characteristics were evaluated among the 89 participants suffering from tinnitus. When exposed to a noisy environment, 54 (60.7%) participants reported tinnitus escalation. Also, 44 participants (49.4%) reported decreased tolerance to noise. Laterality of tinnitus was reported as follows: Left (*n =* 25, 30.1%), Central (*n =* 43, 51.8%) and Right (*n =* 15, 18.1%). Concerning the type of tinnitus, narrow band noise was reported by 33 (40.2%) participants with tinnitus, while 49 (59.8%) reported the pure noise type of tinnitus. Tinnitus evaluation results according to THI are shown in Table 5.

Tinnitus pitch was also evaluated for this group of participants. Results can be found in the Appendix A.

Hearing thresholds at frequencies ranging from 250 Hz to 16,000 Hz were evaluated in both ears for all 122 participants. The mean hearing threshold was calculated for both groups, Tinnitus and No Tinnitus, and compared using the Welch two sample t-test considering the average of both ears (Table 6), for each participant and each frequency. The results for the worse ear and the best ear can be found in the Appendix A, respectively).

The mean hearing thresholds were higher in the Tinnitus group when compared to the No Tinnitus group, for all frequencies and all assessments (worse ear, best ear and average of both ears). In all cases, significant differences were found for hearing thresholds ranging from 2000 Hz to 10,000 Hz.

Results from the speech audiometry recognition (STR) pointed in the same direction—participants suffering from tinnitus performed significantly worse both in the left ear (mean difference 3.1, *p* = 0.03) and the right ear (mean difference = 3.7, *p* < 0.01).

The presence of tinnitus was found to be significantly associated with worse hearing for both PTA and HFA, as the dependent variables in a multiple linear regression model, after controlling for age and sex. In the presence of tinnitus, PTA was higher by 4.75 dB (*p* = 0.008), and HFA was higher by 9.47 dB (*p* < 0.001). Suffering from tinnitus did not seem to significantly worsen VHFA (*p* = 0.201), although VHFA 4.24 dB was worse on average in the tinnitus group. In all the three models, being a man and being older were found to be significantly associated with increasing HL (Table 7).

In our data, the presence of tinnitus does not appear to be associated with age or gender. Welch two sample t-test comparing the mean age in both groups, Fisher exact test and chi-square test used to compare the distribution along the age groups and gender, respectively, all revealed no sign of association of these variables with the presence of tinnitus. Still, controlling for age and gender, the association between the presence of tinnitus and HL at different frequencies was assessed through logistic regression models; higher values of PTA and HFA were all found to be significantly associated with an increased risk of tinnitus (*p* < 0.01 in both cases). The odds of tinnitus were estimated to increase by a factor of 2.07 (95%CI: 1.23–3.7) and a factor of 1.85 (95%CI: 1.30–2.77) for each 10dB increment in PTA and HFA, respectively. As for VHFA, although higher HL in these frequencies was found to be associated to the presence of tinnitus, this was not statistically significant (*p* = 0.199).

### 3.1. Tinnitus and QoL

The effect of tinnitus severity on QoL was firstly assessed using MOS SF-36 physical and mental summary scales, respectively PCS and MCS. For this purpose, one-way ANOVA models were used. In both cases a significant effect of tinnitus severity was found (F = 4.809, *p* = 0.01 for PCS and F = 5.777, *p* = 0.004 for MCS). Pairwise comparisons between the levels of tinnitus severity concerning PCS revealed no significant difference between the group with no tinnitus and the group suffering from Irrelevant/Mild tinnitus (*p* = 0.92), but there were significant differences between the group with no tinnitus and the Moderate/Severe/Catastrophic (*p* = 0.019) and between the group suffering from Irrelevant/Mild and the group with Moderate/Severe/Catastrophic tinnitus (*p* = 0.022). Similar results were found concerning MCS. For the pairwise comparisons listed above, the *p*-values were 0.453, 0.004 and 0.040, respectively (*p*-values adjusted by the Tukey method for comparing a family of three estimates). Given these findings, post-hoc analyses proceeded to evaluate the effect of the tinnitus severity on the subscales of the MOS SF-36 questionnaire.

Age and gender were identified as potential confounders in the analysis assessing the relation between presence/severity of tinnitus and QoL. The Pearson correlation between the MOS SF-36 scores and age was calculated and tested for statistical significance. All correlation values were negative, pointing towards the idea that QoL tends to decrease with age. Statistically significant values of correlation were found between age and MOS.GHP (*p* = 0.021) and MOS.RE (*p* < 0.001). Concerning gender, men systematically scored higher than women, except on MOS.GHP and MOS.HC where, on average, women scored slightly higher than men (not statistically significant, *p* = 0.599 and *p* = 0.292 respectively). Men’s scores were significantly higher than women’s scores on the MOS.PF (*p* = 0.009), MOS.RP (*p* < 0.001), MOS.BP (*p* < 0.001), MOS.SF (*p* = 0.042), MOS.RE (*p* = 0.017) and MOS.PCS (*p* = 0.001).

The domains of MOS SF-36 were assessed for their association with tinnitus severity. Mean scores for the three groups (No Tinnitus, Irrelevant/Mild Tinnitus and Moderate/Severe/Catastrophic Tinnitus) were compared controlling for age and gender as these variables can be significantly associated with MOS SF-36 results, as seen above.

Since the several dimensions of the MOS SF-36 questionnaire cannot be considered independently from one another, the relation between the scores and tinnitus severity was assessed via a multivariate linear regression model where all dimensions of MOS SF-36, except MOS.HC (due to its different registry scale) were considered as the dependent variables and age, gender and tinnitus severity entered in the model as independent variables. Pairwise comparisons between the levels of the tinnitus severity variable were performed after fitting the model, with *p*-value adjusted by Tukey method for comparing a family of estimates. Means and standard deviations for the MOS SF-36 scores can be found in Table 8. Most of the MOS SF-36 domain scores worsened as the severity of tinnitus increased. The p-values of the significant differences of such means at level α = 0.05 and at level α = 0.1 (for trend identification), controlling by age and gender, after multiple testing correction, can be found in the Appendix A. Significant differences in pairwise comparisons are also identified in Table 8. Despite smaller score values of MOS SF-36 dimensions in the group with Irrelevant/Mild Tinnitus when compared to the group with No Tinnitus, such differences were not statistically significant. However, significant differences were found between the No Tinnitus group and the Moderate/Severe/Catastrophic Tinnitus group, as well as between the two groups within the subjects with Tinnitus, regarding General Health Perception and almost all scales of the emotional component of the MOS SF-36.

A graphical representation of the mean scores for the three groups, and the corresponding 95% confidence interval bands, can be found in the Appendix A.

### 3.2. Tinnitus and Psychological Distress

The effect of tinnitus severity on psychological symptoms was first analysed through a one-way ANOVA considering the general severity index of distress, GSI. A significant effect of the severity of tinnitus was found (F = 4.89, *p* = 0.009). Pairwise comparisons between the levels of tinnitus severity concerning GSI revealed no significant difference between the group with no tinnitus and the group suffering from Irrelevant/Mild tinnitus (*p* = 0.273) as well as no significant difference between the group with no tinnitus and the Moderate/Severe/Catastrophic (*p* = 0.361). A statistically significant difference was, however, found between the group suffering from Irrelevant/Mild and the group with Moderate/Severe/Catastrophic tinnitus (*p* = 0.007) (*p*-values adjusted by the Tukey method for comparing a family of three estimates).

Associations between specific BSI scores and general scores on BSI.PST and BSI.PSDI, age, and gender were evaluated to investigate whether age and gender act as confounders in the assessment of the effect of tinnitus severity on the results of BSI questionnaire. Age was found to be significantly associated with some of the BSI scores, namely BSI.DEP (r = 0.18, *p* = 0.047), BSI.SOM (r = 0.17, *p* = 0.067) and BSI.PHOB (r = 0.16, *p* = 0.078). As for gender, average values of the BSI scores were compared between sexes by Welch two sample *t*-test. No significant differences were found (all *p*-values > 0.13).

Means and standard deviations for the BSI scores can be found in Table 9. The relation between the BSI scores and tinnitus severity was assessed via a multivariate linear regression model, controlling for age. Pairwise comparisons between the three levels of tinnitus severity were performed after fitting the model, with p-value adjusted by Tukey method for comparing a family of estimates. Significant differences (*p* < 0.05) and the *p*-values of the significant differences of such means at level α = 0.05 and at level α = 0.1 (for trend identification), controlling for age, after multiple testing correction, can be found in the Appendix A. Significant differences in pairwise comparisons are also identified in Table 9. A graphical representation of the mean scores for the three groups, and the correspondent 95% confidence interval bands, can be found in the Appendix A.

## 4. Discussion

In the present study we aimed to identify aspects that can contribute to the diagnosis and can guide therapeutic interventions for tinnitus patients. We explored clinical, and sociodemographic characteristics, psychological symptoms, and QoL in a sample of Portuguese participants (*n =* 122) aged between 55 and 75 years. All participants were distributed into a “No Tinnitus” (*n =* 33) or a “Tinnitus” (*n =* 89) group according to the absence or the presence of this symptom. The two groups were found to be homogenous since no significant differences were found between them regarding age, sex, marital status, exposure to noise and hearing difficulties. No significant differences were found between these groups regarding the studied comorbidities (diabetes, cardiovascular disease, high blood pressure, hypercholesterolemia, thyroid, measles, meningitis, mumps, tuberculosis, ear diseases), hormone therapy, ototoxic medication, and smoking habits. The group of “Tinnitus” was further divided into “Irrelevant/Mild” (*n =* 54) and “Moderate/Severe/Catastrophic” (*n =* 35) subgroups for analysis considering the effect of the severity of Tinnitus in QoL and mental health.

In our study the use of antidepressant/anxiolytic medication was significantly associated with tinnitus (*p* = 0.011, Chi-squared test) suggesting that patients suffering from tinnitus present some kind of mental health impairment, which is corroborated by our results with BSI. Our findings support the use of CBT for tinnitus to address negative automatic thoughts, safety behaviors and inaccurate beliefs [6]. Those results strongly propose including psychologists in a multidisciplinary approach, with different professionals involved in tinnitus treatment.

We also found that tinnitus was significantly associated with poorer hearing levels at standard speech frequencies and high frequencies, but not regarding very high frequencies. We attribute this fact to the average age of our participants (mean 63.8 ± 5.7). Studying a younger population (mean age 37.25 ± 10.25) found that participants with high frequency hearing loss were significantly older and had higher scores on the tinnitus questionnaires in comparison to those with normal high frequency audiometry [21]. Regarding tinnitus management, while in younger adults, especially those presenting a normal standard audiometry, it is useful to have the evaluation of extended high frequency audiometry, usually this is not the case for those older than 65, according to our study results. These findings are important contributions for tinnitus subtyping, as in younger adults hearing loss in higher frequencies is likely to play a role in tinnitus pathophysiology, but this effect is probably reduced in older individuals [22]. Several studies have focused on the consequences of hearing on QoL, including in the older adult population with hearing loss [23,24,25]. Specifically, hearing loss is associated with depression and social isolation [26,27]. One of the explanations for this association is the ‘cascade’ hypothesis, where long-term deprivation of auditory input can impact cognition functioning, through impoverished input, or via effects of hearing loss on social isolation and depression [28,29], leading in some cases to the development of cognitive deterioration or dementia [30]. A recent review shows evidence in the association of hearing loss with clinically relevant depression symptoms, indicating that an assessment and intervention of comorbid depression in hearing loss is essential to promote mental well-being among older individuals [31].

The majority of the individuals with tinnitus (*n =* 54, 60.7%), when exposed to noise, reported tinnitus escalation. Almost half of the tinnitus participants (*n =* 44, 49.4%) also reported decreased tolerance to noise. Some considerations could be held regarding possible common underlying mechanisms of tinnitus and sound tolerance such as the central auditory gain [32].

Regarding our findings, lower scores on MOS SF-36 are significantly associated with the escalation of tinnitus severity. Thus, our results show a significant association between having tinnitus and having a decreased QoL, particularly on the mental component of the scale (MCS). Age and sex were found to be associated with QoL and hence they might be confounding factors in the assessment of the effect of tinnitus in the MOS SF-36 results. Including age and sex in our statistical models allowed us to measure the effect of tinnitus severity in QoL after removing the effect of age and sex. These results support the profiling of tinnitus by a standardized QoL instrument [7,8].

Our results also show that moderate to severe tinnitus, classified according to THI score, have a significant impact in QoL scored by MOS SF-36, which reflects the multidimensional attributes of tinnitus-related complaints. When the tinnitus severity increases, there is a decrease in the QoL, both physical and emotional, but mainly regarding to Perception of Health, Social Functioning and Mental Health (comparison between the Irrelevant or Mild Tinnitus and Moderate, Severe or Catastrophic Tinnitus).

Mental health problems are the largest single source of disability, accounting for 23% of the total disease burden. Worldwide, around “450 million people suffer from mental and behavioral disorders. One person in four will develop one or more of these disorders during their lifetime” [33]. Our significant results clearly identify psychopathologies in patients suffering from tinnitus as a major factor. Most individuals suffering from tinnitus become habituated to this symptom [34], and those individuals who become disturbed by tinnitus usually have some additional (psychological or not) comorbidity, for example hyperacusis, vertigo, anxiety, depression, headache, etc., that make the individual more prone to focus attention on tinnitus [12,35,36,37,38].

Different studies have shown that psychological variables play an important role in the perception of tinnitus [39,40]. Our findings regarding BSI reveal that the patients more severely affected by tinnitus (classified by THI) are also those who present more severe psychopathology situations (evaluated through BSI). Concerning the dimensions of the BSI questionnaire, our results show that in most cases no significant differences were found between individuals from No Tinnitus and Irrelevant/Mild Tinnitus groups, with the latter group, surprisingly, performing better than the former. Regarding higher tinnitus severity (Irrelevant/ Mild Tinnitus and the group Moderate/Severe/Catastrophic Tinnitus group), significant differences (*p* < 0.05) were found in the Obsessive–compulsive, Depression, Anxiety, Hostility and Phobic Anxiety scales.

The Obsessive–Compulsive scale focuses on the thoughts, impulses and actions of the individual that are considered irresistible but are of an undesired nature [41]. Most troublesome tinnitus cases arise from the fact that patients focus too much attention on tinnitus, so this would seem likely in those individuals with higher obsessive–compulsive tendency. Moreover, catastrophic and negative thinking may be presented and can be related to higher severity of tinnitus, and consequently lower QoL [6,34,42,43].

Anxiety is the principal psychological symptom in 28% of tinnitus patients with high predictive value for the development of tinnitus-related psychological distress [44,45,46]. The Anxiety Scale evaluated in BSI is described signs such as nervousness and tension, feelings of apprehension, fearful feelings, feeling tense and keyed up, some somatic manifestations, often with panic attacks [41]. On the other hand, tinnitus has been associated with major depression, reported in 48% to 60% of tinnitus sufferers [47,48,49,50]. The literature shows correlation between tinnitus severity and worse stages of depression and anxiety [51,52]. Some patients present phobic disorders and a tendency to social isolation [53]. The pathophysiology involving these psychologic comorbidities and tinnitus, and their mutual influence, are still a matter of debate. In fact, these psychological comorbidities can be pre-existent or induced by tinnitus. These studies support our results, being that depression is described in our study as showing symptoms of dysphoric mood and signs of withdrawal from life interest, lack of motivation, feelings of hopelessness and suicidal ideation [41].

Different explanations have been formulated to understand the relationship between tinnitus and anxiety and depression. From a neurobiological point of view, it is postulated that the generalized activation of the limbic system (responsible for our emotions and behavior) can have a central role in individuals with tinnitus and anxiety, thus establishing a connection with the auditory system [54,55]. The role of cortisol as a mediator of the psychological symptoms related to tinnitus may be another explanation for the association between tinnitus and anxiety, with tinnitus individuals showing high levels of cortisol [56,57]. It is observed that individuals with anxiety and depression also have high levels of cortisol, leading to the possibility of a positive feedback cycle, which may induce the intensification of tinnitus [58].

Study Limitations

Participants in this study did not have a psychologist’s or psychiatric evaluation so we relied entirely on self-reporting to categorize participants as having a psychological complaint or not. Furthermore, the questionnaire used, the BSI, evaluates the emotional status of the previous seven days only so does not capture general longer term psychological complaints which may fluctuate in the same way that tinnitus is known to [59,60].

## 5. Conclusions

An increasingly ageing population worldwide raises concerns about QoL, with augmented years lived with disability. Taken together, hearing loss disorders, tinnitus disorders, and accompanying comorbidities have a high negative impact on the QoL of the affected persons, especially if the grade of tinnitus severity is high.

Our study provides evidence for tinnitus negatively impacting the QoL in older individuals (aged from 55 to 75). Thus, patients with Moderate/Severe/Catastrophic Tinnitus present significantly lower QoL.

The severity of tinnitus was associated with the severity of psychopathological symptoms evaluated with BSI. In fact, patients from the Moderate/Severe/Catastrophic Tinnitus group, exhibit significant differences in Depression, Anxiety, Hostility and Phobic Anxiety scales compared to tinnitus-free individuals.

Our study brings new insights concerning the importance of the holistic assessment and management of the individual relevant to tinnitus as a multidimensional symptom [7].

Accordingly, assessing individuals with tinnitus and therapeutic strategies should be multidisciplinary to ensure coverage of all dimensions of the tinnitus patient. Moreover, therapies should be personalized to the patient, after proper information and with respect for choice and individual needs.

## Figures and Tables

**Table 1 brainsci-11-00953-t001:** Sociodemographic characteristics.

	All (*n =* 122)		No Tinnitus (*n =* 33)		Tinnitus (*n =* 89)	
Sociodemographic Characteristics	*n*	%	*n*	%	*n*	%
Age group (years)						
55–59	33	27.0	9	27.3	24	27.0
60–64	40	32.8	12	36.4	28	31.5
65–69	25	20.5	5	15.2	20	22.5
70–75	24	19.7	7	21.2	17	19.1
Sex						
Women	63	51.6	18	54.5	45	50.6
Men	59	48.4	15	45.5	44	49.4
Marital Status						
Married	93	76.2	29	87.9	64	71.9
Single/divorced	21	17.2	4	12.1	17	19.1
Widower	8	6.6	0	0.0	8	9.0

*n*: total number of individuals; %: percentage of individuals.

**Table 2 brainsci-11-00953-t002:** Noise related characteristics.

	All (*n =* 122)		No Tinnitus (*n =* 33)		Tinnitus (*n =* 89)	
Noise Related Characteristics	*n*	%	*n*	%	*n*	%
Exposure to noise						
No	85	69.7	26	78.8	59	66.3
Yes	37	30.3	7	21.2	30	33.7
Difficulty in hearing with noise						
No	61	50.0	16	48.5	45	50.6
Yes	61	50.0	17	51.5	44	49.4

*n*: total number of individuals; %: percentage of individuals.

**Table 3 brainsci-11-00953-t003:** Medication related characteristics.

	All (*n =* 122)		No Tinnitus (*n =* 33)		Tinnitus (*n =* 89)	
Medication	*n*	%	*n*	%	*n*	%
Hormone therapy						
No	89	73.0	26	78.8	63	70.8
Yes	33	27.0	7	21.2	26	29.2
Ototoxic medication						
No	81	69.8	23	74.2	58	68.2
Yes	35	30.2	8	25.8	27	31.8
Past antidepressant/anxiolytic						
No	100	82.0	29	87.9	71	79.8
Yes	22	18.0	4	12.1	18	20.2
Present antidepressant/anxiolytic						
No	103	84.4	32	97.0	71	79.8
Yes	19	15.6	1	0.0	18	20.2
Smoking status						
Nonsmoker	69	56.6	21	63.6	48	53.9
Smoker	53	43.4	12	36.4	41	46.1

*n*: total number of individuals; %: percentage of individuals, p-value estimated with Chi-squared test.

**Table 4 brainsci-11-00953-t004:** History of comorbidities.

	All (*n =* 122)		No Tinnitus (*n =* 33)		Tinnitus (*n =* 89)	
Comorbidities History	*n*	%	*n*	%	*n*	%
Cardiovascular Disease						
No	117	95.9	33	100	84	94.4
Yes	5	4.1	0	0	5	5.6
High Blood Pressure						
No	70	57.4	18	54.5	52	58.4
Yes	52	42.6	15	45.5	37	41.6
Diabetes						
No	107	87.7	26	78.8	81	91.0
Yes	15	12.3	7	21.2	8	9.0
Hypercholesterolemia						
No	50	41.0	14	42.4	36	40.4
Yes	72	59.0	19	57.6	53	59.6
Thyroid						
No	107	87.7	28	84.8	79	88.8
Yes	15	12.3	5	15.2	10	11.2
Measles						
No	35	28.7	5	15.2	30	33.7
Yes	87	71.3	28	84.8	59	66.3
Meningitis						
No	118	96.7	33	100	85	95.5
Yes	4	3.3	0	0	4	4.5
Mumps						
No	59	48.4	16	48.5	43	48.3
Yes	63	51.6	17	51.5	46	51.7
Tuberculosis						
No	119	97.5	33	100	86	96.6
Yes	3	2.5	0	0	3	3.4
Ear diseases						
No	116	95.1	31	93.9	85	95.5
Yes	6	4.9	2	6.1	4	4.5
Ear surgery						
No	120	98.4	33	100	87	97.8
Yes	2	1.6	0	0	2	2.2
Cancer						
No	114	93.4	32	97.0	82	92.1
Yes	8	6.6	1	3.0	7	7.9

*n*: total number of individuals; %: percentage of individuals.

**Table 5 brainsci-11-00953-t005:** Distribution of the participants along the THI categories.

	No Tinnitus	Irrelevant	Mild	Moderate	Severe	Catastrophic
*n*	33	17	37	22	12	1
%	27.0	13.9	30.3	18.0	9.8	0.8

*n*: total number of individuals; %: percentage of individuals.

**Table 6 brainsci-11-00953-t006:** Mean and standard deviation of hearing thresholds for the average of both ears.

Frequency Hz	Tinnitus Mean (SD)	No Tinnitus Mean (SD)	Mean Difference	*p*-Value
250	13.4 (7.0)	12.6 (5.1)	0.8	0.481
500	13.9 (6.5)	12.0 (5.4)	1.8	0.120
1000	16.0 (8.4)	13.6 (6.4)	2.4	0.097
2000	23.5 (13.2)	17.8 (8.3)	5.7	0.005 **
4000	40.0 (19.3)	29.5 (15.5)	10.4	0.003 **
8000	53.7 (21.7)	39.1 (20.7)	14.6	0.001 **
10,000	65.1 (21.4)	55.2 (21.7)	9.9	0.029 *
12,000	72.4 (19.1)	71.7 (18.3)	1.7	0.652
14,000	84.8 (20.7)	85.5 (18.4)	−0.7	0.852
16,000	89.3 (25.5)	89.8 (22.9)	−0.6	0.908

* *p*-values < 0.05, ** *p*-values < 0.01, according to Welch two sample *t*-test.

**Table 7 brainsci-11-00953-t007:** Results from the multiple regression models.

	PTA		HFA		VHFA	
	β	*p*-Value	β	*p*-Value	β	*p*-Value
Age	0.44	0.002	1.06	<0.001	1.37	<0.001
Gender M	4.51	0.005	9.88	<0.001	6.86	0.041
Tinnitus Yes	4.75	0.008	9.47	<0.001	4.24	0.201

Dependent variables were PTA, HFA and VHFA. Regression coefficients (β) and corresponding *p*-values are presented for each equation of the multiple regression models. M: men; PTA = average of hearing thresholds at frequencies (500 Hz, 1000 Hz, 2000 Hz and 4000 Hz); HFA = average of hearing thresholds at frequencies (2000 Hz, 4000 Hz and 8000 Hz); VHFA = average of hearing thresholds at frequencies (8000 Hz, 10,000 Hz, 12,000 Hz, 14,000 Hz and 16,000 Hz).

**Table 8 brainsci-11-00953-t008:** Mean and standard deviation of MOS SF-36 scores for the three groups of tinnitus severity and results of pairwise comparison.

MOS SF-36 Domains	No Tinnitus (*n =* 33) Mean (SD)	Irrelevant/Mild (*n =* 54) Mean (SD)	Moderate/Severe/Catastrophic (*n =* 35) Mean (SD)
Physical Functioning MOS.PF	82.5 (20.2)	83.1 (16.6)	76.7 (18.5)
Role-Physical MOS.RP	83.3 (32.9)	78.3 (35.4)	69.9 (41.6)
Bodily Pain MOS.BP	72.7 (21.6)	73.2 (21.5)	61.3 (22.2)
General Health Perception MOS.GHP	66.3 (21.8)	63.9 (18.4)	50.5 (16.3) ^a,b^
Vitality MOS.V	71.7 (21.3)	63.1 (18.4)	57.5 (19.4) ^a^
Social Functioning MOS.SF	83.0 (17.9)	87.7 (14.7)	77.2 (22.0) ^b^
Role-Emotional MOS.RE	86.8 (27.6)	79.9 (33.5)	73.6 (37.4)
Mental Health Index MOS.MH	80.9 (18.5)	74.6 (18.7)	61.8 (19.7) ^a,b^
MOS.HC	3.1 (0.6)	3.0 (0.7)	3.4 (0.9)

^a^ *p* < 0.05 significant differences between No Tinnitus and Irrelevant/Mild Tinnitus group; ^b^ *p* < 0.05 significant differences between Irrelevant/Mild Tinnitus group and Moderate/Severe/Catastrophic Tinnitus group.

**Table 9 brainsci-11-00953-t009:** Mean and standard deviation of BSI scores for the three groups of Tinnitus severity and results of pairwise comparison.

BSI Scales	No Tinnitus (*n =* 33) Mean (SD)	Irrelevant/Mild (*n =* 54) Mean (SD)	Moderate/Severe/Catastrophic (*n =* 35) Mean (SD)
Somatization BSI.SOM	39.2 (27.1)	39.7 (22.4)	51.6 (18.4)
Obsessive–compulsive BSI.O-C	46.4 (20.2)	43.2 (17.1)	53.1 (15.1) ^b^
Interpersonal Sensitivity BSI.I-S	40.6 (26.8)	37.8 (23.1)	42.5 (25.6)
Depression BSI.DEP	27.9 (28.8)	33.5 (24.4)	47.3 (23.6) ^a,b^
Anxiety BSI.ANX	37.2 (24.7)	39.8 (20.2)	51.3 (19.8) ^a,b^
Hostility BSI.HOS	34.5 (25.8)	37.1 (21.5)	49.9 (15.5) ^a,b^
Phobic Anxiety BSI.PHOB	24.2 (29.1)	27.5 (28.5)	43.2 (16.9) ^a,b^
Paranoid Ideation BSI.PAR	37.5 (28.1)	41.0 (22.3)	43.9 (25.1)
Psychoticism BSI.PSY	21.2 (29.4)	23.6 (27.7)	33.7 (24.0)
Positive Symptom Total BSI.PST	52.3 (11.4)	43.0 (17.2)	54.2 (14.1) ^b^
Positive Symptom Distress Index BSI.PSDI	44.4 (13.8)	40.4 (14.0)	47.8 (12.8) ^b^

^a^ *p* < 0.05 significant differences between No Tinnitus and Irrelevant/Mild Tinnitus group; ^b^ *p* < 0.05 significant differences between Irrelevant/Mild Tinnitus group and Moderate/Severe/Catastrophic Tinnitus group.

## Data Availability

The availability data is in the body of manuscript or in the Appendix A.

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
