# Peer review of "Quality of Life and Psychological Distress in Portuguese Older Individuals with Tinnitus"

_brainsci, 2021, doi:10.3390/brainsci11070953_

Round 1
Reviewer 1 Report
Some more detailed comments on the possible relationship between depression and social funcioning should be added. Is there a possible relationship) Which presumably come first? Some important evidence are present in audiological literature.
Author Response
Comment from Reviewer1.: Some more detailed comments on the possible relationship between depression and social funcioning should be added. Is there a possible relationship) Which presumably come first? Some important evidence are present in audiological literature.
Thank you for your suggestion, in fact the psychological comorbidities can be pre-existent or induced by tinnitus. So, in some cases it’s really difficult to understand which comes first and this is a very interesting topic for discussion.
Please find in line 785: ‘Several studies have focused on the consequences of hearing on QoL, including in the older adult population with hearing loss [23-25]. Specifically, hearing loss is associated with depression and social isolation [26,27]. One of the explanations for this association is the ‘cascade’ hypothesis, where long-term deprivation of auditory input can impact cognition functioning, through impoverished input, or via effects of hearing loss on social isolation and depression [28,29], leading in some cases to the development of cognitive deterioration or dementia [30]. A recent review shows evidence in the association of hearing loss with clinically relevant depression symptoms, indicating that an assessment and intervention of comorbid depression in hearing loss is essential to promote mental well-being among older individuals [31].’ And on line 1044: ‘On the other hand, tinnitus has been associated to major depression, being reported in 48% to 60% of tinnitus sufferers [47-50]. The literature shows correlation among tinnitus severity and worse stages of depression and anxiety [51,52]. Some patients present phobic disorders and tendency for social isolation [53]. The pathophysiology involving these psychologic comorbidities and tinnitus and their mutual influence are still a matter of debate. In fact, the psychological comorbidities can be pre-existent or induced by tinnitus.’
At subchapter of conclusions we rephrased at line 1072: ‘Taken together, hearing loss disorders, tinnitus and accompanying comorbidities have a high negative impact on the quality of life of the affected persons especially if the grade of tinnitus severity is high.’

Reviewer 2 Report
The manuscript presents a good quality work regarding the quality of life of older population with tinnitus compared with "no tinnitus" controls.
Please state the question regarding the tinnitus status.
Please give more explanation about the exclusion criteria of participants younger than 55.
Author Response
Comments from Reviewer 2.: Please state the question regarding the tinnitus status.
Thank you for pointing out this aspect. In order to answer your question we introduced in line 250: ‘Were included participants with subjective chronic and non-pulsatile tinnitus (duration longer than six months). So, we excluded objective or somatosensory tinnitus.’
Please give more explanation about the exclusion criteria of participants younger than 55.
In order to better explain the choice of age ranging of our study population we introduced new paragraphs at line 203: ‘WHO classifies aging into four stages: Middle age: 45 to 59 years; Elderly: 60 to 74 years old; Elder: 75 to 90 years old; Extreme old age: 90 years onwards.
Since the focus our study was on older individuals, we decided that the ranging from 55 to 75 years old would give us a good appreciation of aging process in regards of tinnitus and related comorbidities. Our sample was gender balanced (63 women and 59 men).’
We also reviewed the manuscript for checking of spelling incorrections or other mistakes. Please check the version with tracked changes.
